# Detecting Salient Image Objects Using Color Histogram Clustering for Region Granularity

**DOI:** 10.3390/jimaging7090187

**Published:** 2021-09-16

**Authors:** Seena Joseph, Oludayo O. Olugbara

**Affiliations:** Department of Information Technology, Durban University of Technology, Durban 4000, South Africa; seenaj@dut.ac.za

**Keywords:** color contrast, contrast ratio, histogram clustering, region saliency, saliency detection

## Abstract

Salient object detection represents a novel preprocessing stage of many practical image applications in the discipline of computer vision. Saliency detection is generally a complex process to copycat the human vision system in the processing of color images. It is a convoluted process because of the existence of countless properties inherent in color images that can hamper performance. Due to diversified color image properties, a method that is appropriate for one category of images may not necessarily be suitable for others. The selection of image abstraction is a decisive preprocessing step in saliency computation and region-based image abstraction has become popular because of its computational efficiency and robustness. However, the performances of the existing region-based salient object detection methods are extremely hooked on the selection of an optimal region granularity. The incorrect selection of region granularity is potentially prone to under- or over-segmentation of color images, which can lead to a non-uniform highlighting of salient objects. In this study, the method of color histogram clustering was utilized to automatically determine suitable homogenous regions in an image. Region saliency score was computed as a function of color contrast, contrast ratio, spatial feature, and center prior. Morphological operations were ultimately performed to eliminate the undesirable artifacts that may be present at the saliency detection stage. Thus, we have introduced a novel, simple, robust, and computationally efficient color histogram clustering method that agglutinates color contrast, contrast ratio, spatial feature, and center prior for detecting salient objects in color images. Experimental validation with different categories of images selected from eight benchmarked corpora has indicated that the proposed method outperforms 30 bottom-up non-deep learning and seven top-down deep learning salient object detection methods based on the standard performance metrics.

## 1. Introduction

Salient object detection is an arduous open research problem aimed at retrieving the most conspicuous visually distinct foreground information from an image in a manner reminiscent of the human vision system [1,2,3,4,5,6,7,8]. It is a challenging task because human vision is difficult to mimic by automated systems. Salient object detection methods attempt to extract points and regions of a visual scene that are more significant to human visual attention by forming a map that defines how a region stands out from its background and analyzing image surroundings [1,8,9]. Saliency detection is extensively used to mitigate the complexity of image analysis and speed up the processing time, and it has gained popular applications in the disciplines of computer vision and artificial intelligence [8,10]. The numerous application domains of saliency include image segmentation [11,12,13,14], object detection and recognition [15,16,17], anomaly detection [18,19], image retrieval [20,21], image compression [22], object classification [23], object tracking [24], image retargeting, and summarization [25,26], alpha matting [26], target detection [27], video object segmentation [28], video summarization [29], user perceptions of digital video contents [30], and visual tracking [31]. Countless applications of saliency detection have led to the occurrence of numerous methods for saliency computation. The orthodox saliency detection methods can be classified into two approaches, top-down and bottom-up, based on the perspective of information processing [9,32,33,34,35]. The top-down approach is task-driven with semantic information, prior knowledge and it focuses on supervised machine learning from a plethora of training images [8,32,36]. The approach has had great success in salient object detection with the progress of deep learning methods [8,37,38,39,40,41]. Deep saliency detection methods are often trained with a large set of finely annotated pixel-level ground truth images [42,43,44]. However, the performance of deep learning methods is highly dependent on the construction of well-annotated training datasets and can be adversely affected [43,45].

The bottom-up approach is data-driven without semantic information but grounded in the connotation of primitive features such as color, intensity, shape, and texture that are simple to implement [32,46,47]. The bottom-up methods compute uniqueness in primitive features of image pixels and surrounding regions. These saliency detection methods have extensively used different visual rarities to separate foreground and background regions in images. The visual rarities include color prior [3,48,49], contrast prior [32,50], brightness prior [11,51], background prior [33,52], boundary prior [4,53], center prior [13,54], shape prior [55], context prior [25], object position prior [56], and connectivity prior [7,44,53,57]. However, despite the development of several methods for salient object detection, there are still intrinsic challenges with different categories of images. The presence of cluttered and non-homogeneous background regions, inter-object dissimilarity, heterogeneous objects with varying sizes, counts, and positions have led to ambiguous and diverse challenges. Examples of image categories are salient objects with erratic sizes, positions, and counts, cluttered backgrounds, and low dissimilarity among regions of heterogeneous foreground or heterogeneous background. The task of completely highlighting salient objects in different image categories is still not adequately resolved in most of the existing saliency methods [58,59,60]. The other major challenge is the mitigation of computational complexity because salient object detection is an essential preprocessing stage in computer vision.

This study addresses the problem of automatic selection of optimum homogenous regions for image abstraction to reduce the computational complexity, improve the effectiveness, and increase the efficiency of salient objects detection for different classes of images. The method of color histogram-based clustering has been developed in this current study for this purpose. A near resolution of detecting salient objects in different images has been achieved by successfully integrating holistic strategy of color contrast, contrast ratio, center prior, and regional spatial feature while adhering rigidly to the efficacy requirement of salient object detection. The idiosyncratic contributions of this study to the existing research in computer vision are threefold:The comprehensive review of related literature on salient object detection methods and approaches to demonstrate trends, uniqueness, recency, and relevance of the current study.The construction of a novel bottom-up saliency computation method that exploits the strategy of color contrast, contrast ratio, center prior, and spatial feature to obtain a robust salient object detection process.The intensive experimental comparison with different prominent salient object detection methods that were reported in the literature to determine the effectiveness of the proposed method.

The remainder of this paper is succinctly structured as follows. Section 2 gives a comprehensive review of the related literature. Section 3 describes the proposed salient object detection method. Section 4 explicates the intensive experimental comparison of the proposed method against the existing modern methods based on the widely known benchmarked corpora and performance evaluation metrics. Section 5 provides a discussion of experimental results and a brief concluding remark.

## 2. Review of Literature

A plethora of color image saliency detection methods have been reported in the literature, strikingly developed in the last two decades. The bottom-up method by Itti et al. [61] is considered a cornerstone strategy grounded in the biological model for eye-fixation activities of humans. The method is based on center-surroundedness differences in color, intensity, and orientation that can detect spatial discontinuities in a scene. It estimates the locations of visual gaze by computing multi-scale feature maps using a Gaussian pyramid. The second category of saliency detection methods has emerged from the works [15,62] where saliency was defined as a binary segmentation problem. The third wave of saliency has emerged with the introduction of the convolutional neural network (CNN) to lessen the reliance on center bias knowledge. Neurons in the CNN model with large receptive fields of global information can enhance the detection of the most salient region in an image [63]. A plethora of salient region detection methods have been developed among which bottom-up methods are pervasive because of their simplicity, elegance, and computational efficiency.

### 2.1. Bottom-Up Saliency Detection Methods

Bottom-up salient object detection methods are stimulated by the human visual system and can be categorized into the eye fixation prediction (EFP) approach [61,64,65] and salient object detection (SOD) approach [4,6,66,67,68,69,70]. The two approaches are based on the definition of saliency as “where people look” or “which objects stand out” in an image [71]. The former approach focuses on the prediction of a location where people are freely observing natural scenes, while the latter approach targets the detection and segmentation of salient objects in images. The SOD approach has gained more popularity than the EFP approach because of its ability to identify the essential characteristics of salient objects than predicting their locations only [56,72]. The salient regions are usually considered as perceptually distinct image parts that are dissimilar to their backgrounds [42]. The dissimilarity, rarity, or uniqueness has been extensively studied with several advancements in the bottom-up SOD approach [42,60]. The contrast features have gained substantial popularity in SOD applications because they reflect the human visual system that gives more attention to high contrast regions. The contrast-based salient object detection methods are frequently employed locally or globally.

#### 2.1.1. Local Contrast-Based Saliency Detection

Local contrast-based salient object detection methods compare the rarity of image units to their surrounding neighborhoods. The local uniqueness maps from various feature channels are nonlinearly integrated to highlight the most attractive region [64]. The center-surroundedness local contrast by the difference of mean filter applies Euclidean distance between average feature vectors of center and surrounding regions to formulate center-surroundedness color contrast [15]. The center-surroundedness method [73] applied self-resemblance measure to compute pixel-level saliency by employing the local steering kernels and matrix cosine similarity-based nonparametric kernel density to discriminate a pixel from its surroundings. The pixel-wise center-surroundedness local saliency method [74] used a probabilistic framework fused with features of illumination, color contrast, and optical flow to compute pixel saliency. The saliency is computed at pixel level with the help of a sliding window over the entire image to yield stable results on the selected datasets. However, the method has difficulty in clearly separating objects from the background when there is no distinct visual contrast between the object and background pixels.

A local central surroundedness contrast-based saliency map using inverse wavelet transform for each color channel at a multi-spatial scale was proposed [75]. The center-surroundedness method that integrates local color contrast features and center bias to compute saliency exploits sparse sampling and kernel density estimation [54]. The method [36] incorporated compactness cues and local contrast with a diffusion process using manifold ranking to lessen the constraint of local contrast that highlights the boundaries of objects rather than the entire region. However, consideration of local relevance among neighboring regions can lead to incorrect suppression of salient regions, especially in images with heterogeneous salient object features [76]. A local contrast-based method for detecting small targets by computing contrast between the targeted small regions and surrounding regions was proposed [77]. Due to the limited spatial neighborhood consideration in the local contrast method, large salient regions can be easily excluded [42].

#### 2.1.2. Global Contrast-Based Saliency Detection

Global contrast saliency detection methods are capable of evenly highlighting a complete salient region by assigning comparable salient values across similar regions and are extensively used in salient object detection. The frequency-tuned global contrast-based method was introduced to measure pixel-level saliency by computing the Euclidean difference between each pixel feature and the mean color feature in L*a*b* color model of a smoothed image [48]. Color histogram was introduced as a global contrast method that employed the Gaussian mixture model (GMM) to define a weighted sum of the color difference of region contrast to the rest of image regions [50]. It was hypothesized that high contrast to a neighborhood region exhibits more saliency than high contrast to faraway regions [50]. The spatial relationship of regions was integrated to increase the effect of surrounding regions in saliency computation because the distribution of spatial compactness is an important complementary feature to color contrast [36]. However, regardless of the importance of contrast-based saliency detection, these methods are still prone to some inherent limitations. The global contrast methods alleviate the problem of attenuated object saliency values of local contrast methods, but highlighting salient regions uniformly is still a delinquent they are facing. The incorrect highlighting of background region than the salient object is another drawback of global contrast-based methods, especially for images with complex backgrounds or large salient objects [47].

The global color cues based on statistics and color contrast was recently utilized to overcome the inherent limitation of exploiting surroundedness cue alone [70]. Failure to detect a salient object linked to image borders is a major drawback of this method. The method based on context-aware saliency detection that integrated local and global features was introduced in [25] to obtain a patch-level saliency. A method based on both local and global approaches was presented for saliency computation by Liu and Wang [78]. The authors used local contrast difference features to obtain an attention map based on a block variance map. A learning-based method that combined both global and local saliency features was described in [67]. A method of salient object detection that agglutinated multiscale extrema of local perceptual color difference, global measure rarity, and global center bias was recommended to detect large salient objects [3]. The successful detection of salient objects in images that share similar color contrast features between foreground and background regions is a major drawback that has been identified in a recent method that integrates contrast, background, and foreground features [44].

The methods based on contrast prior generally work well on images with distinctive color contrasts, but have difficulty when there is no distinct visual contrast between the foreground and background regions [78]. Hence, it is vivacious to incorporate useful information on foreground and background regions for segmenting diverse image categories. Suitable prior knowledge can enhance the quality of saliency detection, but the ultimate results are not absolute on images with complex background and foreground objects that possess variable shapes, sizes, locations, and appearances. The center prior methods are not sufficient to trace salient objects when the image background is framed near the image center or salient objects are close to the image boundary [36,47,59,79]. The methods of exploiting background and connectivity priors have suffered from incorrect suppression of salient objects that touch image boundary [6,32,79,80]. Some existing methods are insufficient to detect large salient objects that overlap foreground and background regions because they consider the objects as part of a background and accomplish low accuracy on saliency detection [3]. Color contrast prior is not sufficient to successively detect salient objects from images with low color contrast between foreground or background and complex background or foreground scenes. This restraint emphasizes that even though a significant improvement has been witnessed, salient object detection remains a challenging issue because of image diversity, inherent complexity, and uncertainty of salient regions [32,81,82].

#### 2.1.3. Graph-Based Saliency Detection

The bottom-up saliency detection methods based on graph structure have recently gained attention for object detection. Graph-based methods partitioned an image into regions using the superpixels algorithm. They consider each image region as a graph and nearby nodes are related using weighted edges to diffuse saliency information by seeds and propagation. However, superpixel-based algorithms require the specification of the desired number of superpixels beforehand, but users may not have such knowledge. The graph-based method reported in [83] used boundary prior and manifold ranking to measure the similarity of a region to foreground or background cues. Even though the method has demonstrated good results in terms of computational efficacy, it is still challenged by inaccurate detection of boundary superpixels as background queries. In addition, it is not ideal for detecting salient objects from images with a complex background scene.

The graph-based method proposed in [80] utilized random walk in absorbing Markov chain for salient object detection by exploiting boundary prior. However, boundary-positioned objects and objects that show high color similarity to background regions are challenging cases for this method. A label propagation method using deformed smoothness was developed based on manifold ranking by exploiting objectness and smoothness constraints to overcome the aforementioned limitation [84]. In general, most of the existing graph-based methods are not adequate to successfully separate salient objects from images with complex background scenes or salient objects with various features [8]. The graph-based salient object detection method that integrates background prior and objectness before creating a coarse saliency map was proposed to overcome the deficiencies of graph methods [8]. The authors used the boundary-guided graph-based iterative propagation technique to refine a saliency map. Still, this method has challenges in completely suppressing background noise and successfully highlighting salient objects from complex scenes.

#### 2.1.4. Supervised Learning Saliency Detection

Other saliency detection methods have exploited high-level features through the supervised machine learning approach. The supervised learning methods form regional descriptors by extracting sophisticated image features and regional level saliency scores are predicted by utilizing a classifier or regressor [42]. Kim, Han, Tai, and Kim [67] proposed a learning-based saliency detection method that estimates global saliency using high dimensional color transform and local contrast by regression. A tree-based classifier was used to separate the identified superpixels into the foreground, background, and unknown regions. Saliency maps based on a linear combination of a high dimensional color model and learning-based methods were aggregated to obtain a final saliency map. However, accurate classification of background and foreground regions of images with high foreground and background color similarity is a challenging case for this method. A salient object detection method that used the supervised machine learning approach was proposed to fuse regional descriptors and high-dimensional features [85]. These learning methods have comparatively achieved better performance, but they are still inadequate for rapid and simple detection of salient objects because of the inherent computational time complexity.

### 2.2. Saliency Methods for Challenging Image Categories

The wide spectrum of image datasets with uncertain and diverse salient objects can be more challenging for the existing saliency detection methods. There are few methods proposed to address salient object detection on a few challenging image categories. In [85], a supervised learning method was developed to detect salient objects that are farther from the image center but located at the image boundary. A pixel-based center-surroundedness method was proposed to detect salient objects and multiple salient objects from complex scenes [86]. A learning method based on logistic regression was proposed in [87] to detect a complex salient object by deriving saliency from ultra-contrast features. A saliency method that utilized multiscale extrema of local perceptual color difference was devised to successfully detect large salient objects [3]. A saliency detection method that applied deformed smoothness-based manifold ranking was presented to overcome the problem of misclassified salient objects with low contrast backgrounds [84]. A saliency detection method based on the fusion of foreground-center with background priors was recently proposed to solve the challenge of detecting salient objects touching image boundary [68]. The color volume of regions was created by the superpixels algorithm [88] with perceptual homogenous color differences between regions exploited to detect salient objects.

A graph-based method based on global and local cues that integrated background and foreground saliency maps were introduced to overcome the inadequacy of existing graph-based methods in successfully detecting salient objects from complex scenes [89]. The detection of salient objects adjacent to the image boundary is a major glitch for methods that treat boundary regions as background [6]. The glitch was addressed by a graph-based saliency detection method that exploited background divergence using edge weight and center prior [6]. These various contributions have emphasized the development of myriads of saliency detection methods to address some of the challenging image categories. However, a single method that can be used for a wide gamut of image categories is still far away from a breakthrough in object detection research.

### 2.3. Deep Learning Saliency Detection Methods

Deep-learning-based methods are leading the league of top-down salient object detection methods. A saliency detection method reported in [90] aggregated deep neural network (DNN) sparse and dense labeling schemes to extract hybrid image features by multiscale kernels. A DNN that embedded high-level features captured using the CNN, contrast, and spatial information-based low-level features for detecting saliency were proposed [91]. A deep network saliency prediction method that exploited the in-network feature hierarchy of CNN and stochastic gradient descent (SGD) for training was proposed in [38]. A data-driven deep-learning-based saliency detection method utilizing semantic features of salient objects based on a fully convolutional neural network (FCNN) and non-linear regression to refine a saliency map was proposed in [92]. A multi-context deep learning method that integrated global and local contexts based on CNN was proposed in [93]. The use of semantic information and prior knowledge of a scene has helped to achieve superior performance by these learning methods, but the feat comes with the superfluous cost of the computational complexity of training and testing [45]. The demand for large-labeled datasets for saliency detection is a strenuous chore and deep learning methods generally require high-performance computing devices for training and testing that generally refrained them from real-time applications [41,94].

### 2.4. Unit of Processing

Bottom-up saliency detection methods are primarily characterized by low-level features and computational efficiency [42]. The methods usually consider either individual pixels or regions of pixels as the unit of processing [35,95]. The abstraction of an image into pixel regions has a significant role in reducing computation time by considering each region as a unit of processing. Hence, the selection of an image abstraction process is a crucial step for computationally efficient bottom-up saliency methods.

#### 2.4.1. Pixel-Based Saliency Detection

Pixels are considered in pixel-based methods as independent image elements for extracting features. Pixel-based saliency detection methods are computationally expensive and they disregard pixel connectivity and structure of regions that influence pixel saliency values [56]. The early methods [61,64,74,86] can be classified as the pixel-based approach. However, object interior suppression, boundary-blurring, and poor object segmentation are their main shortcomings. Hence, pixel-based methods are seldom explored in recent times [96]. In contrast, the region-based methods cluster an image into an abstract representation of homogenous regions and perform saliency computation by contrasting region pairs [56,66,95]. They consider non-overlapping patches or homogenous regions as image elements for saliency computation.

#### 2.4.2. Region-Based Saliency Detection

The string of different methods has been applied in the literature to construct a group of pixels that is more attracted by salient regions than individual pixels [46,66]. The methods include fixed-size patches or blocks [25,97,98,99,100] graph-based segmentation [66,85], mean-shift [101] and simple linear iterative clustering (SLIC) superpixels algorithm [4,7,36,68,79,102]. A Bayesian framework was developed to integrate bottom-up saliency and top-down knowledge for saliency computation by extracting features of patches [100]. Regions were computed as non-overlapping patches by dividing an input image into patches of pixel size [97]. The dissimilarity of patches is calculated in terms of spatial distance, center bias, and reduced dimensional space to compute saliency. Since statistical features of patches are irregular, both background and foreground objects can be presented in regular patches, but these methods tend to produce fuzzy salient maps. A patch-based saliency detection method was proposed to combine both local and global features for computing rarity-based saliency [99]. Multiscale patches were used to compute a saliency map that integrated context prior, center prior, local, and global features [25]. Regional covariance color, orientation, and spatial features were employed to obtain structural information of image patches for saliency computation [98]. The method in [101] was aimed at resolving the issues related to patch-based methods by employing a mean-shift clustering algorithm to segment an image into uniform regions of non-overlapping patches. The saliency of a patch was computed by integrating local, global, and spatial features.

The histogram of color namespaces was utilized to measure color differences for computing the weighted attention saliency maps [70]. The saliency detection method described in [66] applied a graph-based segmentation algorithm to construct uniform regions that can preserve object boundaries more efficiently. A learning-based saliency detection method reported in [85] used a graph-based segmentation to divide images into regions to compute region-level saliency. In addition, the global contrast-based saliency detection method reported in [50] used graph-based segmentation to divide an input image into regions. However, the efficiencies of these methods are limited because of the computational complexity of a graph-based region creation process [103]. The connotation of superpixels was introduced as an alternative method for dividing an image into perceptually homogenous regions [104]. In recent times, myriads of salient object detection methods have employed the superpixels approach to divide the input images into perceptually homogenous regions. However, the main limitation of these methods is that isolated or cluttered pixels cannot be grouped correctly because of the constraint of spatial domain connectivity coupled with the determination of an optimum number of superpixels [56]. Moreover, the counts of small and large superpixels may, respectively, lead to under-segmentation and over-segmentation of images, which can lead to the non-uniform highlighting of salient regions [95]. The impact of superpixels granularity on the performance of saliency detection was demonstrated in [68]. Hence, the accuracy of saliency detection is highly dependent on the optimal selection of the superpixels granularity [105]. Determining an optimal superpixels granularity is a difficult task because of the diverse image categories. Multi-level abstraction of an input image by repeatedly applying the superpixels algorithm was proposed to obtain the finest and coarsest abstraction of regions to resolve the granularity problem of superpixels [56]. However, the iteration process increases the computational complexity that can adversely affect the performance of the saliency detection process in real-time applications.

In summary, emphasizing high-contrast edges while suppressing the interior of salient regions is a major obstacle of the pixel or patch-based methods. Region-based image abstractions are considered superior to pixel- or patch-based methods because they can employ a richer feature representation for saliency detection. Superpixel-based methods have gained popularity in recent years because of their computational efficiency. However, finding an optimum superpixels granularity is a challenging task for the superpixel-based image abstraction process. This is because the efficiency and robustness hallmarks of saliency detection methods are highly dependent on the granularity of superpixels. This empathizes the significance of constructing an efficient method that can automatically detect the number of regions for image abstraction. The proposed color histogram-based image abstraction can automatically detect the appropriate image region granularity based on the color distribution of an image as explicated in the subsequent section.

## 3. Methods

The novel regional color histogram clustering method is introduced in this study for detecting salient objects in red, green, and blue (RGB) images. The quantized RGB (QRGB) color image is the input to the histogram-based clustering process to reduce the number of colors in the input image. The numerous color models used in saliency detection methods include RGB [67,70,106], hue, saturation, value (HSV) [107], lightness, redness, yellowness (L*a*b*) [66,68,84,108,109] and combination of color models [67,85,106]. This study has used the QRGB color image for clustering while the L*a*b* color image was applied for the extraction of color features because of its perceptual uniformity [50,66,110]. Literature has shown that color quantization in the RGB color model relatively performed better than quantization in the L*a*b* color model [111]. The purpose of transforming the original RGB color image into the L*a*b* color image instead of the QRGB image was to minimize the effects of quantization error. Consequently, the L*a*b* color model was selected for color feature extraction in the range of [0, 1] to suppress the effect of any possible dominant colors and to take the intrinsic advantages of perceptual uniformity of the color model [108,112]. The proposed method exploits the strategy of color contrast, contrast ratio, spatial feature, and center prior to efficiently compute pixel-level saliency scores. The method is comprised of three essential steps of input image segmentation into regions, calculation of region saliency scores, and post-processing of the computed saliency map. The outline of the proposed method for salient objects detection is depicted in Figure 1.

### 3.1. Segmentation of Input Image

The segmentation of an image into regions of similar pixels is an acceptable preprocessing stage in a saliency detection process. The purpose is to reduce the computational complexity of image data because pixels in a region exhibit similar color features [32]. Multilevel image segmentation methods such as superpixel-based clustering, K-means clustering, and mean-shift clustering have been extensively utilized to divide an image into multiple regions. The superpixel-based segmentation is extensively used among these methods [4,8,36,68,79,80,102,106,113,114,115]. Nevertheless, superpixel-based segmentation methods have suffered from high computational complexity because of multiple iterations and they are not adequate for diversified classes of images [68,116]. The automatic detection of region count is a difficult problem because of the diversity in color images. Moreover, the number of homogenous regions in an image is unknown. The regional color histogram clustering proposed in this study was inspired by the properties of a color histogram to obtain pixel regions. Color histogram is widely used in computer vision algorithms because it can provide the global statistics of color images to describe the proportion of different color features [117]. The segmentation method is achieved in two subprocesses of color quantization and region generation.

#### 3.1.1. Color Quantization

The true-color image contains a maximum possibility of $256^{3}=16,777,216$ colors that is generally greater than the number of pixels in an image [118,119]. Since extremely rare colors are not significant for highlighting salient regions, less dominant colors can be excluded for saliency detection [66]. Color quantization is a widely used technique for merging less dominant colors into dominant colors to significantly reduce the computational complexity of image processing [119,120]. The minimum variance method [66] or pixel intensity clustering algorithm [121] can be effectively applied to perform color quantization. However, the ‘imquantize’ built-in color quantization function in MATLAB (2019a, The MathWorks, Inc., Natick, MA, USA) was effectively used to obtain the dominant colors of the input RGB image. The function uses the multilevel image thresholding method of Otsu to quantize an input image into the specified number of desired colors. The individual color channels of red (R), green (G), and blue (B) of the RGB color model was quantized into QR, QG, and QB at the level of 8 to realize a maximum number of 512 colors. This number corresponds to a maximum of 512 possible regions in a color image. The quantized intensity levels are combined to obtain the index, $Q_{RGB}$ of a quantized RGB color in the color palette using Equation (1).
(1)$$\begin{array}{l} Q_{RGB}=w_{r}*Q_{r}+w_{g}*Q_{g}+w_{b}*Q_{b}\\ \; \end{array}$$
where $w_{r}=8$, $w_{g}=64,$ and $w_{b}=1$ are the weights of R, G, and B colors, respectively. The green channel was assigned the highest weight value because the human visual system is highly sensitive to the green color than other colors [122].

#### 3.1.2. Region Generation

The automatic generation of regions is based on a global color histogram with ${q*\;q*q}_{}$ bins computed from the QRGB image using all pixels in the input image. The 8 × 8 × 8 quantization system with 512 bins is ideal by considering a tradeoff between performance and computational complexity [123]. This line of reasoning has been followed to accept parameter ‘q=8’ to be sufficient for effective color quantization. The global histogram was used to create regional clustering for image abstraction. The histogram bins with pixels are used as representative regions and bins that have no pixels are discarded. Thus, a data structure with ‘M’ entries is created to store features of pixels that fall into each region. This technique implies that each region is represented by a feature vector that includes the average color pixel intensity, average color pixel coordinates, and distance from the regional center to the image screen center.

### 3.2. Calculation of Region Saliency

The global color contrast $CC(r_{i})$ of a region $r_{i}$ is determined in terms of the region weight $W_{i}$ and color difference of a region to all other regions in the image as in Equation (2).
(2)$$CC(r_{i})=\sum_{j=1}^{M}W_{j}{\left\|\left(L_{i},a_{i},b_{i})-(L_{j},a_{j},b_{j}\right)\right\|}_{2}$$
where $\left(L_{},a_{},b_{}\right)$ is the color value of the region in L*a*b* color model, $\|\cdot {\|}_{2}$ indicates the $L_{2}$ norm, and M is the number of regions automatically detected. There will be a maximum of 8 colors in an image, assuming each image channel has 2 distinct intensity levels. The number, M of the possible colors or regions in a quantized color image, will lie in the range of [8, 512]. The regional weight function $W=(W_{1},\ldots ,W_{M})$ is integrated into the region saliency calculation process. The weight function will account for the contribution of high saliency by larger regions than for the smaller ones. The weight of a region is calculated based on the relative probability of the pixels in the region to emphasize the color contrast of larger regions [50,66], as defined by Equation (3).
(3)$$W_{i}=\frac{f_{i}}{f}$$
where $f_{i}$ is the frequency of the pixels occupied in each region $r_{i}$ and f is the total number of pixels in the input image. The spatial contrast function $\mathrm{SC}(r_{i})$ integrates the global color contrast with the spatial feature and color ratio of a region $\left(r_{i}\right)$ as follows:(4)$$\mathrm{SC}(r_{i})=W_{i}CC(r_{i})+\sum_{j=1}^{M}W_{j}\phi (r_{i},r_{j})\exp(-\mathrm{DS}(r_{i},r_{j}))$$

In a divergence from the work [66] that utilized regional saliency differences as a weighting coefficient to suppress the effect of non-salient regions, our method utilizes a more resilient function, $\phi (r_{i},r_{j})$ based on the contrast ratio given by Equation (5). The contrast ratio is an important aspect of image quality that measures the difference between the maximum and minimum brightness of an image. In the context of this study, it measures the difference between the maximum and minimum brightness of regions in an image.
(5)$$\phi (r_{i},r_{j})=\left(\frac{CC(r_{i})+0.05}{CC(r_{j})+0.05}\right)$$

The significance of center prior in saliency detection as given by Equation (6) has been highlighted in literature following the fundamental assumption that salient objects are framed near the image center while background pixels are distributed at the image borders [36,59,66,68,102]. It is usually formulated and extensively used in literature as a Gaussian distribution [3,36,66,106,113]. The region saliency score $CS(r_{i})$ is obtained in terms of spatially weighted color contrast and the Euclidean distance between the region spatial center and image screen center. This is to integrate the center prior with color contrast, contrast ratio, and spatial feature using Equation (6).
(6)$$CS(r_{i})=SC(r_{i})*\exp(-DS(S(r_{i}),C)/\alpha^{2})$$
where $S(r_{i})$ is the spatial center of a region and C = (0.5, 0.5) is the image screen center. Since salient objects are always not positioned at the image center, the concept of center prior can lead to the exclusion of salient objects located at the image boundary or inclusion of a background region [66,106,124]. This can occur, especially when an object possesses multiple colors such that object colors at the image center are different from those in the background. The parameter α∈[0.1, 1.0] is incorporated into the region saliency score function to strengthen the center prior. Even though the function can compute a low saliency score for a region around the boundary, an appropriate α value can make adequate salient objects more salient, regardless of their positions. In addition to the color contrast features, spatial features play a significant role in human attention, and the use of spatial coherence in saliency computation is widely accepted by many researchers [36,50,66,67,68,97]. The spatial distance $DS(r_{i},r_{j})$ between two regions is computed using Equation (7).
(7)$$DS(r_{i},r_{j})={\|\left(C_{i}^{x},C_{i}^{y}\right)-\left(C_{j}^{x},C_{j}^{y}\right)\|}_{2}$$
where $\left(C_{i}^{x},C_{i}^{y}\right)$ is the spatial center of a region $r_{i}$ that is computed by averaging the x and y coordinates of pixels in the region. The regional saliency score generated is normalized to the range of [0, 1] before assigning the pixel level saliency. The saliency score of each pixel is assigned by the saliency score of the respective region to obtain the saliency map, $C_{Map}$ as shown in Equation (8). The assignment is based on the assertation that pixels belonging to the same region have the same saliency.
(8)$$C_{Map}=CS\left(r_{i}\right)$$

### 3.3. Post-Processing of Saliency Map

The post-processing stage is performed to eliminate undesirable artifacts that may be present at the saliency detection stage because of the quantization error. In our method, post-processing is accomplished by three stages of morphological reconstruction, mean suppression, and nonlinear intensity mapping. Morphological reconstruction is a good approach to retrieve objects with connected components of similar intensity values while keeping information such as contour, shape, and intensity to suppress background noise concomitantly [125,126]. Inspired by this, our method adopted a grayscale morphological reconstruction with a disk-shaped structuring element of radius, to uniformly highlight the detected saliency regions while effectively suppressing background noise. Morphological reconstruction can also suppress high-intensity values of salient objects by leaving unobtrusive background regions with non-black pixels [70]. Background noise is further suppressed by computing the average intensity value of the reconstructed saliency map and then subtract this value from each pixel intensity of the saliency map to overcome the pitfall of reconstruction [127,128]. The final saliency map, $S_{map}$ of the intensity values of salient objects, is adjusted to an appropriate range of intensity values by a nonlinear mapping introduced in [70]. The mapping is as given by Equation (9), where values for the parameters r and γ are 0.02 and 1.5, respectively.
(9)$$S_{map}=Map(C_{map},r,\gamma )$$

These three post-processing stages have effectively facilitated the suppression of background noise that is presented in the initial saliency map to obtain the final desirable saliency map.

## 4. Experimental Results

This study has applied the properties of salient objects to categorize various images into different groups to provide a more comprehensive experimental evaluation of the proposed saliency detection method. Figure 2 shows these properties to be the location of salient objects (center or boundary), object sizes (salient objects that overlap center and boundary regions), number of salient objects (multiple objects), color contrast (low contrast), and complex background. The performance of the proposed method was validated against 30 modern bottom-up and seven deep-learning-based top-down methods. Since we do not have access to source codes of the deep learning, and five bottom-up saliency methods, they were considered for the extended complex scene saliency dataset (ECSSD). The rest of the methods were included for comparison on six categories of images. The only parameter that was used in the proposed method is the central bias weight, α, selected experimentally as α∈[0.1, 1.0].

### 4.1. Datasets

Experimental images were selected from different benchmarked datasets of MSRA10K [62], ASD [48], SED2 [129], ImgSal [130], DUT OMRON [83], ECSSD [131], HKU IS [132], and SOC [133]. These datasets have been extensively used for evaluating salient object detection methods [5,38,56,63,66,68,79,80,106,133]. The MSRA10K is a descendent of the Microsoft Research Asia (MSRA) dataset, where many images in this dataset are often with a single salient object and simple background [26,57]. The ASD is a subset of the MSRA dataset with ground truth region annotation, single foreground, and simple background [35]. These two datasets are mainly used for selecting center located, boundary located, foreground or background overlapped, and low contrast images. The SED2, ImgSal, and DUT-OMRON datasets are known for multiple salient objects with relatively complex backgrounds [57,102,112,134]. The images with multiple salient objects and complex backgrounds were selected from these three datasets. The salient objects of the SED2 dataset exhibit different color, position, and size properties.

In addition, images from the ECSSD were selected for evaluation [131]. The ECSSD dataset includes 1000 images that contain salient objects with colors that are affected by background regions, and salient objects with heterogeneous colors, sizes, and location properties to present huge ambiguity for the methods of salient object detection. The dataset is fundamentally considered to be complex for performance comparisons [5,57,67,68,84,135,136]. The HKU-IS dataset has 4447 complex scenes with multiple disconnected objects that are highly similar to the background regions with a diverse spatial distribution [43,132]. Salient objects in clutter (SOC) is a recently introduced dataset [133] and is a subset of the common objects in contexts (COCO) dataset [137]. SOC is a challenging dataset of salient objects with attributes reflecting occlusion, cluttered background, and challenges in real-world scenes developed for evaluating CNN-based salient object detection methods. The proposed method is not featured for detecting salient objects from occluded or cluttered backgrounds and is not based on the CNN approach. It was tested against 1500 images of the SOC dataset to determine its ability to detect salient objects in real-world scenes.

### 4.2. Methods Compared

This study has tried to incorporate a combination of different bottom-up saliency methods of pixels, and regions based on different approaches such as center-surroundedness, global contrast-based, graph-based, learning-based, and prior knowledge, as shown in Table 1. Many of these bottom-up saliency detection methods have been typically benchmarked in several studies [9,10,63,66,70,84,112,138]. In addition, we have included the RPC [66], and CNS [70] methods because of their relatedness to our method, which is not related to top-down methods. However, we have also compared our method with seven deep-learning-based top-down methods because they are central to a lot of high-end innovations in recent times.

**Table 1 jimaging-07-00187-t001:** Saliency methods compared.

Bottom-Up Saliency Methods			
No	Method	Approach and Prior Knowledge	Unit of Processing
1	FES [54]	Center-surroundedness contrast, center prior	Pixel
2	IT [61]	Center-surroundedness, intensity, color, and orientation contrast	
3	GB [64]	Graph-based, center-surroundedness activation map	
4	SeR [73]	Local steering kernel features and color features	
5	SEG [74]	Local feature contrast, boundary prior	
6	SR [139]	Spectral residual approach	
7	AC [15]	Center surroundedness color contrast prior	Patch/Block
8	CA [25]	Global, and local features, context prior, center Prior	
9	SWD [97]	Center prior, color dissimilarity, spatial distance	
10	COV [98]	Local color contrast, center prior	
11	SUN [100]	The local intensity and color features, feature space	
12	MRBF [7]	Boundary connectivity, foreground prior	Region by SLIC algorithm
13	DCLC [36]	Diffusion-based using manifold ranking, compactness local contrast, center prior	
14	MCVS [44]	Background prior, foreground prior, and contrast features	
15	CSV [56]	Global color spatial distribution, object position prior	
16	HDCT [67]	Learning-based approach, global and local color contrast features, location, histogram, texture, and shape features	
17	FCB [68]	Foreground and background cues, center prior	
18	MC [80]	Boundary prior, graph-based, Markov random walk	
19	MR [83]	Boundary prior, graph-based manifold ranking	
20	DGL [84]	Graph-based, boundary prior	
21	FBSS [94]	Boundary, texture, color, and contrast priors	
22	DSR [106]	Background prior	
23	MAP [108]	Boundary prior, graph-based, Markov absorption probabilities	
24	BGFG [109]	Background and foreground prior	
25	GR [113]	Convex-hull-based center prior, contrast and smoothness prior, graph-based	
26	BPFS [140]	Global color contrast, background prior, and foreground seeds	
27	RPC [66]	Color contrast, center prior	Regions by graph-based segmentation
28	DRFI [85]	Color and texture contrast features, backgrounds features	
29	CNS [70]	Surroundedness and global color contrast cues	Regional histogram of color name space)
30	SIM [75]	Center surroundedness color contrast	Spatial scale
31	OURs	Color contrast, contrast ratio, spatial feature, and central prior	Regional color histogram clustering
Deep-learning-based top-down saliency methods			
1	MSNSD [38]		
2	MSNSD-A [38]		
3	TSL [90]		
4	LCNN [91]		
5	DS [92]		
6	MCDL [93]		
7	[141]		

In this study, we run the source codes of the methods of AC, BGFG, CNS, DCLC, DGL, DRFI, GB, GMR, HDCT, IT, MAP, MR, and RPC with their default parameters. The implementations of salient object detection methods in [63] with default parameters were employed to obtain the saliency maps of CA, COV, DSR, FES, GR, MC, SEG, SeR, SR, SUN, and SWD. Since we have no access to the source codes of the remaining methods, they were excluded for qualitative comparison, analysis of computational time complexity, and could not compare with all the selected image categories. The method of FCB was considered for the category of overlap images and ECSSD dataset based on the saliency results provided by their authors.

### 4.3. Evaluation Metrics

The visual observance of saliency maps against the ground truth annotation is generally accomplished by qualitative analysis, which has assisted to scrutinize the degree of resemblance of saliency maps with the ground truth. In addition, a quantitative evaluation was performed to compare the competency of the proposed method against a set of modern methods. The quantitative evaluation is more accurate than the qualitative evaluation that is highly subjective. The standard performance metrics universally used for evaluating salient object detection methods are precision, recall, F-measure, mean absolute error (MAE), and overlapping ratio (OR) [36,63,67,68,70,84,142,143]. Hence, this study has incorporated these metrics to evaluate the performance of the proposed method against the selected modern methods. Precision is the ratio of the number of correctly identified salient pixels to the total number of pixels in a salient map [13,144,145]. Recall or sensitivity is the degree of correctly identified salient regions to the total number of salient pixels in the ground truth [13,144,145].

Precision and recall values were obtained by comparing the binary map equivalent of a salient map with the ground truth image. A fixated threshold value from 0 to 255 was used to bipartite the saliency map to obtain the binary map equivalent. The pair of precision and recall values were computed for each threshold to plot the performances at different situations [63,143]. The successfully identified non-salient pixels are not considered either by precision or recall. This affects the methods that correctly identified non-salient pixels but failed to correctly detect salient pixels [146,147]. In this study, MAE between saliency map and ground truth was also computed for a balanced evaluation to take this effect into account. The MAE is a common metric to measure dissimilarity between the estimated and actual values [26]. It is defined as the average absolute error between the continuous saliency (CS) map, and ground truth (GT). The OR is the ratio of overlapping between the binarized saliency map and ground truth [57], where better performance is indicated by higher values of OR.

### 4.4. Qualitative Results

The visual comparison of our method against the selected existing methods on different categories of images is demonstrated in Figure 3. The ability of the proposed method to effectively suppress non-salient pixels while highlighting the salient objects with well-formed edges regardless of the type of images is perceptible. The proposed method can uniformly and accurately detect salient regions in diverse classes of images over many of the existing modern methods. It is clear from the results that most of the existing methods are performing well on the categories of relatively simple images with single or homogenous objects. However, they present challenges on image categories with complex backgrounds, low contrast, or multiples objects. Methods that utilized boundary prior or background prior such as BGFG, DGL, DSR, MC, MR, and MAP are not able to detect or uniformly highlight objects that touch image boundary as observed in Figure 3b. In contrast, our method has successfully detected and uniformly highlighted the salient objects that touch the image boundaries. This shows the ability of the proposed method in suppressing the background adequately and highlighting salient objects with well-formed edges regardless of the locations of salient regions in images.

Methods that exploited center prior performed well on images with centrally located salient objects, but they showed challenges in some cases such as Figure 3(c2), where two objects (bowl and strawberry) are centrally located. Hence, these methods tend to concomitantly detect both objects as a salient region. However, methods that incorporate color contrast such as RPC, DCLC, GR, CNS, and HDCT have managed to highlight real salient objects. Methods such as MC, MR, MAP, and DGL that exploited boundary prior have failed to detect salient objects in this category of images because they considered black boundary regions as background regions and incorrectly highlighted the white bowl as a salient object. The performance of the proposed method for this category of images is highly commendable because it has demonstrated strength in detecting salient objects, irrespective of variation in sizes as in Figure 3(c1) (small salient object) and Figure 3(c3) (large salient object). The salient object in Figure 3(c3) also shows heterogeneous properties in terms of color and appearance; hence, many modern methods such as DCLC, GR, MAP, MC, MR, and RPC have failed to uniformly highlight salient regions. These methods have managed to highlight only a portion of salient regions rather than the entire salient regions. In contrast, the proposed method shows impressive results that are almost like the ground truth images. The visual results of the existing methods in a complex background are shown in Figure 3d. The results show the power of the proposed method in detecting salient objects from complex and heterogeneous backgrounds while all other methods show lower performance. The performance of DRFI is comparatively better than the rest of the existing methods because of the inclusion of color and texture features along with the use of multi-level pre-segmentation maps to detect multi-invariant objects.

Salient objects with low contrast to the background are considered a challenging case for contrast-based and graph-based methods. The visual representation of this image category is demonstrated in Figure 3e; it is worth noticing that the performance of all methods except ours is not remarkable. The DGL method proposed a deformed smoothness constraint to overcome this challenge of graph-based methods. However, DGL still had failure cases as in Figure 3(e3) that it cannot effectively handle low contrast objects. The result in Figure 3(e3) shows that the performance of DRFI is not free from the limitation of contrast-based cues because of the use of feature extraction by contrast vectors. The performance of DSR is relatively better than the rest of the methods; nevertheless, the results are not free from background noise as in Figure 3(e3). The regional contrast-based method of RPC based on low-level color contrast features also demonstrated poor performance on low contrast objects. The proposed method has illustrated good results as compared to the listed modern methods. The ability to uniformly highlight salient objects in the category of multiple objects is still challenging for many of the modern methods because of the heterogeneous nature of objects as illustrated in Figure 3f.

The results of Figure 3(f1) have illustrated that many methods such as CNS, DGL, DSR, MAP, MC, and MR can detect only one object. The proposed method has again demonstrated its ability in detecting heterogeneous objects from this class of images. Except for the proposed method, only DRFI shows relatively better results for this category of images. The images that belong to the overlapped category are generally larger and they touch the image boundary and image center as shown in Figure 3g. The proposed method shows an outstanding performance for images in this category like the previous categories. Moreover, the graph-based methods or diffusion-based methods such as DGL, MR, MC, and MAP have achieved good performance on the category of overlapped images. In opposition to the performance of DSR for the category of low contrast images, DSR has demonstrated poor performance on overlapped images because the method has incorrectly assigned all image boundaries as a background template. The methods such as COV, FES, IT, GB, SeR, SUN, SWD, and SIM, as illustrated in Figure 3, generally showed challenges in highlighting salient objects from all the listed categories of images.

The ECSSD dataset is generally well-known for salient objects with heterogeneous properties and occluded backgrounds. The proposed method has again demonstrated remarkable results on images from this dataset. The learning-based methods such as HDCT and DRFI have shown better performance on images in this dataset. The results indicate the merits of the proposed method on a wide spectrum of image categories and obviously, its output is more reliable with results that are almost like the ground truth in comparison to the existing modern methods.

Figure 4 shows the qualitative results of the proposed method in comparison with the top-performing methods on the challenging HKU-IS and SOC datasets. The HKU-IS is well-known for multiple and disconnected salient objects that show high similarity to the background regions. The SOC dataset contains images that are closer to real-world conditions. The qualitative results are shown in Figure 4 highlight the performances of the proposed method and six other methods that generally perform well for the category of multiple objects. The proposed method shows good results on these two challenging datasets with the output almost resembling the ground truth. In Figure 4b, for instance, the proposed method highlighted the salient object as in the ground truth image, while other methods detected all objects on the table. Similarly, the performance of the proposed method is commendable, regardless of the complexity of these images (Figure 4a,c,d).

### 4.5. Quantitative Results

The quantitative comparison of the proposed method against other methods in terms of the metrics of precision, recall, F-measure, MAE, and OR are revealed in Table 2 to objectively reinforce the performance of the proposed method on diverse categories of images.

#### 4.5.1. Salient Objects Located at Image Boundary

Table 2 and Table 3 show comprehensive results of the investigated methods based on the standard performance metrics. The results show that our method scored the highest precision (0.945), F-measure (0.932), and OR (0.844) with a slightly lower recall as compared to the learning-based methods of HDCT and DRFI. In terms of MAE, the proposed method achieved the second-best score of 0.062, where DCLC and DSR recorded the best score of 0.057. In addition to our method, the performances of DCLC and GR are perceptible. The GR used a convex hull to estimate salient objects and centroid of the convex hull as center prior instead of image center to favor the detection of salient objects located farther from the image center. The DCLC ranked saliency based on foreground seeds obtained by local contrast and performed well in this category unlike other diffusion-based methods such as MC and MR, which considered the nodes that touch the image boundaries as background seeds. The SeR achieved the lowest precision of 0.532 and for all other metrics, SIM showed the lowest performance. Regardless of the use of a center prior, appropriate selection of α value has enabled the proposed method to produce a robust detection of salient objects located far off the image center. Figure 5 demonstrates the average precision, recall, F-measure, MAE, and OR on the category of boundary images for all the investigated methods.

#### 4.5.2. Salient Objects Located at Image Center

The proposed method achieved the highest precision (0.949), F-measure (0.934), and OR (0.846) for this category of images. The DRFI achieved the best recall score, CNS scored the lowest MAE score of 0.058, followed by MR and DSR with scores of 0.061 and 0.062, respectively, while the proposed method scored 0.067. The DGL shows improvement in terms of F-measure and OR on the center category of images than the boundary category of images, with SIM being the last. This image category is relatively simpler as objects are located far away from the image boundary and located close to the image center to favor methods that exploit the location prior. The performances of center prior-based methods such as FES, COV, and SWD are relatively better than those of the boundary images. The methods such as DCLC, DGL, GR, and MR show precision values between 0.9 and 1.0, while the average F-measure, MAE, and OR have demonstrated the superiority of the proposed method over the comparative methods, as depicted in Figure 6.

#### 4.5.3. Salient Objects with Complex Background

The results achieved by the investigated salient object detection methods indicated that performance is generally challenging for this image category. However, the proposed method shows its capability for precisely detecting salient objects, and it is the only method that recorded a precision score between 0.900 and 1.000 with the highest F-measure of 0.885. The proposed method also achieved the best MAE score of 0.120. Surprisingly, DCLC gave good results for boundary and center image categories but achieved unsatisfactory results for this category of images. In contrast, DGL and DRFI improved their performances for this category of images. The deformed smoothness constraint-based manifold ranking approach used by the DGL method has helped to improve performance for this image category compared to other manifold ranking-based methods such as MR. As stated in [7], results obtained for MR have demonstrated poor performance on complex background images when compared to other categories of images. The SIM again scored the lowest performance on this category of images. Figure 7 shows the average precision, recall, F-measure, MAE, and OR for all the investigated methods. The results show the capability of the proposed method in the handling of images with a complex background to exhibit its superiority over the other methods investigated.

#### 4.5.4. Salient Objects with Low Color Contrast to Background

The proposed method showed strength in effectively detecting salient objects from the low contrast object category like other image categories. It achieved the highest scores for most performance metrics, except for the recall. The highest recall values on images from this category are between 0.7 and 0.8 while the proposed method scored a recall value of 0.715. It is evident from this research that the existing methods investigated have difficulty in effectively detecting salient regions when an object shares a similar color contrast with background regions. This includes learning-based methods because the performances of HDCT and DRFI are not encouraging on images from this category. Furthermore, contrast prior-based methods such as DCLC, GR, CNS, and RPC have demonstrated the lowest performances when compared to other categories of images. Like the results of other categories, SIM again scored the lowest values for all the performance metrics. Figure 8 shows the average precision, recall, F-measure, MAE, and OR of all methods, wherein the capability of the proposed method in the handling of salient objects with low color contrast to the background is superior to the existing methods investigated.

#### 4.5.5. Multiple Salient Objects

It is hard to detect salient objects when they exhibit heterogeneous features in terms of location, color, size, and count. This image category contains multiple objects with varying locations, sizes, counts, and colors. However, the performance of the proposed method is commendable with the best value for precision (0.876), F-measure (0.853), MAE (0.836), and OR (0.695). The learning-based methods of DRFI (0.818) and HDCT (0.791) scored the highest recall value, followed by the proposed method (0.786). The results obtained by the rest of the methods clearly showed difficulty in detecting multiple salient objects with heterogeneous properties. In this category of images, all methods showed relatively poorer performance in terms of MAE.

In addition to our method, DGL showed comparatively good results with the second-highest values for F-measure (0.834) and OR (0.656). The limitation of COV in detecting multiple salient objects is clear from these results as it shows a comparatively low performance when compared to other image categories. This is because of the consideration of the assumption of spatial coincidence in multiscale saliency computation [98]. The SIM method again scored the lowest performance on this category of images. Figure 9 shows the average precision, recall, F-measure, MAE, and OR of all the investigated methods on image category of multiple objects. The results show the capability of the proposed method in handling salient objects with heterogeneous properties in terms of position, count, and size.

#### 4.5.6. Images with Foreground and Background Overlapped Objects

The average precision, recall, F-measure, MAE, and OR scores achieved for the category of overlapped images are illustrated in Figure 10. In this category of images, the DCLC obtained the best overall performance with the highest recall (0.804), OR (0.790), and F-measure (0.934). The proposed method achieved the highest precision value of 0.986 and is highly competitive with DCLC. Surprisingly, the graph-based methods of DGL (0.105) and MR (0.114) achieved the best MAE scores, while SIM and SUN scored inferior MAE values of 0.389 and 0.326, respectively. In this category of images also, the SIM method recorded the lowest performance.

#### 4.5.7. Comparison with ECSSD Dataset

The results of the proposed method were further compared against all the 30 bottom-up saliency methods on the ECSSD dataset as in Table 3 and Figure 11 to evaluate its performance. The ECSSD dataset is well known for harboring complex images while the superiority of the proposed method is obvious because it has achieved the best values of precision (0.853), F-measure (0.790), MAE (0.163), and OR (0.573). The learning-based method of DRFI and graph-based methods of DGL, FBSS, and MRBF also achieved better results; however, only the proposed method managed to score precision above 0.800. The foreground and backgrounds seed selection methods such as MRBF and FBSS have also achieved a better MAE score compared to BGFG, which is also based on background and foreground seed selection. The DCLC that showed superiority in the image category of overlap declined its performance on the ECSSD dataset. The SIM method showed the lowest value for most of the performance metrics, except the MAE, while the method of SUN scored relatively the worst value for MAE. The effectiveness of the proposed method in detecting salient objects from a wide range of image categories has been successfully proven by experiments.

#### 4.5.8. Comparison with Deep-Learning-based Top-down Saliency Methods

The proposed method is not related to top-down or deep-learning-based methods. However, we have extended the quantitative comparison to seven deep-learning-based top-down saliency detection methods on the ECSSD dataset to demonstrate the superiority of the proposed method. Recently, the performance of deep-learning-based top-down methods brought some challenges for bottom-up saliency methods [140]. However, the performance of our method has revealed the ability of bottom-up saliency detection methods can compete favorably with deep-learning-based top-down methods. Table 4 illustrates the comparison of our method with deep learning methods based on F-measure and MAE values reported in the original references. Regardless of the complex nature of the ECSSD dataset, the proposed method has achieved the best F-measure (0.790) when compared to deep-learning-based methods. In terms of MAE, the deep learning method of DS shows a relatively best value of 0.160, but the MAE value of the proposed method is 0.163, which is a very close result. This result shows that the proposed method is even competitive with deep-learning-based top-down methods. The F-measure and MAE scores in Table 3 and Table 4 illustrate that deep-learning-based methods of MSNSD-A and MSNSD, respectively, scored the second and third best F-measure values and higher than those of other bottom-up methods, including the graph-based and learning-based methods listed in Table 3. In terms of MAE scores, deep learning methods of DS and LCNN scored the best values and showed that their saliency maps are close to the ground truth. However, the performances of these methods are highly dependent on supervised learning based on labeled training data [44]. Due to the high dependency and sensitivity of deep learning methods on training datasets, these methods are restricted from using real-time and diverse categories of images [42,94].

#### 4.5.9. Comparison with HKU-IS and SOC Datasets

Table 5 summarizes the performances measured by precision, recall, F-measure, MAE, and OR of the investigated methods on HKU-IS and SOC datasets. In the comparison based on these two datasets, we have considered the methods of DCLC, DGL, and DRFI because they showed comparatively good performances on all the selected categories of images. These methods are among the top-performing methods, especially for the category of images with multiple objects, and HKU-IS is well-known for images with multiple salient objects. In addition, three deep learning methods of MSNSD-A, MSNSD, and MCDL were included for comparison on HKU-IS in terms of the F-measure and MAE scores reported in the original references. We excluded all other deep learning methods for comparison on the SOC dataset because of inaccessibility to their source codes. The deep learning methods of MSNSD-A and MSNSD scored the highest F-measure (0.837) and lowest MAE (0.071), and the second-highest F-measure (0.776) was achieved by the proposed method on the HKU-IS dataset. Moreover, our method recorded the best performance in terms of precision (0.813) and OR (0.578). In general, all methods showed weak performance on the SOC dataset. It was recorded in the literature that existing saliency detection methods generally showed unsatisfactory performance with a lower F-measure below 0.45 on realistic scenes with occluded and cluttered backgrounds [133]. It is clear from the experimental results of this study that the performances of the investigated methods decline on this dataset. Surprisingly, our method scored the F-measure of 0.618 where DCLC, DGL, and DRFI scored 0.543, 0.552, and 0.561, respectively. In addition, our method comparatively scored the best value for MAE (0.202), and OR (0.389).

#### 4.5.10. Computational Time Analysis

Salient object detection should mitigate the computational complexity of image analysis by efficaciously detecting regions of interest. Since it is an intelligent pre-processing stage of computer vision tasks, fast and effective detection of the most salient regions is paramount. Computational complexity is a limiting factor of most methods in real-time applications. Deep-learning-based methods are intrinsically suffering from this limitation because of their computational complexity. This study incorporates runtime computational analysis to experimentally demonstrate the efficiency of the proposed method. In the comparison, we had to exclude few methods from running time analysis because of a lack of access to their source codes. The experiment was performed using a machine with an Intel(R) Core (TM) i7-8650U CPU @ 1.90GHz 2.11 GHz, and 8 GB random access memory. Table 6 summarizes the running times of 25 methods on the ECSSD dataset. The proposed method ran much faster than most of the other methods, except MAP, FES, SR, and SWD. It is well illustrated in quantitative and qualitative analysis that FES, SR, and SWD have shown poorer performance, irrespective of computational efficiency. The methods such as DCLC, DGL, and DRFI that are competitive with our method are computationally complex than the proposed method. The CA suffered from high computational complexity and is mainly because of the application of the K-nearest neighbor algorithm to locate the nearest patches. The classical learning-based method of HDCT and DRFI is also computationally expensive because they have consumed more time in feature extraction. The running time of the recent method of CNS is also higher and it is mainly influenced by the sample size parameter used in attention map computation. The DGL is computationally more expensive than other graph-based methods such as GR, MAP, MC, and MR.

## 5. Discussion and Conclusions

### 5.1. Discussion

The proposed method always consistently scored the best performance in terms of precision and F-measure across all categories of images, while the MAE and OR values are always in the top three positions, as illustrated in Table 2, Table 3 and Table 4. The pixel-based methods of GB, IT, SeR, and SR dropped their precision values across the categories of boundary and center objects with a surge in the recall value. The supervised learning methods of DRFI and HDCT scored very high recall values across many of the image categories, but at the cost of low precision and F-measure. Similarly, the method of BPFS scored the highest recall value across images from the ECSSD dataset, but at the cost of low precision and low F-measure. The MAP method recorded some better recall values, but the generalized initial saliency map depends on the Markov absorption probability. This can cause challenges in detecting images that touch boundaries, and it is obvious from the experiments that MAP did not achieve good recall results for the categories of boundary, overlap, and multiple objects. The recall metric is generally not considered a good choice for evaluation because its high value can be the result of highlighting the entire image region. However, the proposed method scored more balanced precision and recall values while at the same time managed to score the highest F-measure on ECSSD, boundary, center, complex background, low contrast, and multiple objects. It has achieved the second-best value for the category of overlapped images. In terms of MAE, the method of CNS showed good performance on a few image categories, such as center, complex background, and ECSSD dataset. However, CNS has failed to achieve the best MAE on the boundary, overlapped, and low contrast images because of the consideration of low-level features such as color and surroundedness cues [70]. In addition, the DSR method consistently achieved lower MAE for all image categories, except for the category of multiple objects and the ECSSD. Superior methods that exploited the principle of center prior can exclude salient objects that touch the image boundary because salient objects are not always located at the image center [47]. However, the proposed method still managed to demonstrate outstanding performance on boundary images with the proper integration of color contrast, contrast ratio, spatial features, and center prior.

The effectiveness of the DGL method in handling various categories of images is higher when compared to other graph-based methods, but at the cost of computational complexity. The run time analysis has demonstrated that DGL is computationally complex than the graph-based methods of GB, GR, MC, MR, and MAP. The GR method has introduced a convex-hull-based center bias to mitigate the common limitation of the center prior map that incorrectly suppresses the salient objects far from the image center. The convex-hull center prior has improved the accuracy of salient objects that touch the image boundary, but this method did not perform well for objects that are positioned at the image center. The DRFI method used a 35-dimensional feature vector that includes geometric, appearance, color, texture, and background features for region description. These features along with multi-level segmentation have led the DRFI method to achieve a good performance on many categories of images. The results computed by the method are free from the limitations of contrast-based methods, regardless of the use of color contrast features. However, the assumption of a narrow image border as a pseudo background can affect the performance of DRFI on the category of boundary images. Computational complexity is another intrinsic drawback of this method. The methods such as DGL, MR, MAP, and MC that exploited the boundary prior have shown relatively low performance on the category of boundary images when compared to the category of images with center prior. This shows the major challenge of boundary prior in treating boundary regions as backgrounds and is not effective when salient objects are near to the image boundary.

The methods such as CNS, DCLC, RPC, and GR that exploited contrast prior have demonstrated relatively low performance on the category of low contrast images. This is because contrast prior works well with images that have distinct color contrast differences between foreground and background regions. This indicates that performances of the investigated methods are highly dependent on salient object properties such as count, location, size, color contrast, or background complexity. However, the proposed method has performed well on most categories of images, irrespective of the various object properties and background complexity. The extended evaluation of the proposed method on HKU-IS and SOC datasets has further revealed the strength of our method in handling images from differing datasets. However, the performances of our method and other bottom-up methods in detecting the salient objects in the cluttered and occluded background were not achieved with remarkable results. This is because the primitive image features such as color, contrast, and texture are not adequate to detect the salient objects from cluttered and occluded images in a meaningful manner [148]. The detection of objects from the cluttered and occluded background can be enhanced by incorporating high-level features [148,149].

The integration of color contrast, contrast ratio, spatial feature, and center prior information in the proposed method has provided adequate segregation of salient regions from non-salient regions and uniformly highlighted salient objects. The accomplishment of the proposed method makes it nearly universal for detecting salient objects in a wide spectrum of images. Moreover, the quantitative comparison of the investigated methods has exhibited the superiority of the proposed method and we were flabbergasted by the performance of our method against the deep-learning-based top-down methods. Finally, all region-based methods have shown good performances when compared to the patch and pixel-wise methods. However, the performances of these methods are completely dependent on the selection of region granularity. Due to the ability of the proposed method to automatically detect the optimum number of regions, it has achieved the best results when compared to the investigated methods. There is always a tradeoff between computational complexity and accuracy. However, this is not the case with the proposed method because we have achieved the best performance while upholding an efficient run time of 0.23 s per image as demonstrated in Table 6. It should be observed that preprocessing was not considered in the proposed as in the case of most methods and can be optional.

### 5.2. Conclusions

This study has enriched the research on salient object detection by proposing a simple, effective, and efficient method that incorporates histogram-based region formation for image abstraction. The method has successfully integrated color contrast, contrast ratio, spatial features, and center prior for achieving an impressive salient object detection process. The method is capable of accurate and robust detection of salient objects from a wide gamut of challenging images by uniformly highlighting. This accomplishment is achieved by the successful integration of color contrast, contrast ratio, spatial feature, and center prior. Experiments on different image categories have established that our method has outperformed all 30 bottom-up saliency methods and seven deep-learning-based top-down saliency methods. The computational efficiency of our method has demonstrated that it can be exploited in real-time applications such as object segmentation and object recognition. The proposed method has proven to be effective and efficient for a large set of image categories, regardless of heterogeneous properties of salient objects, and complex backgrounds. The future work will incorporate texture features and high-level features to improve the detection of salient objects in cluttered and occluded images.

## Figures and Tables

**Figure 1 jimaging-07-00187-f001:**
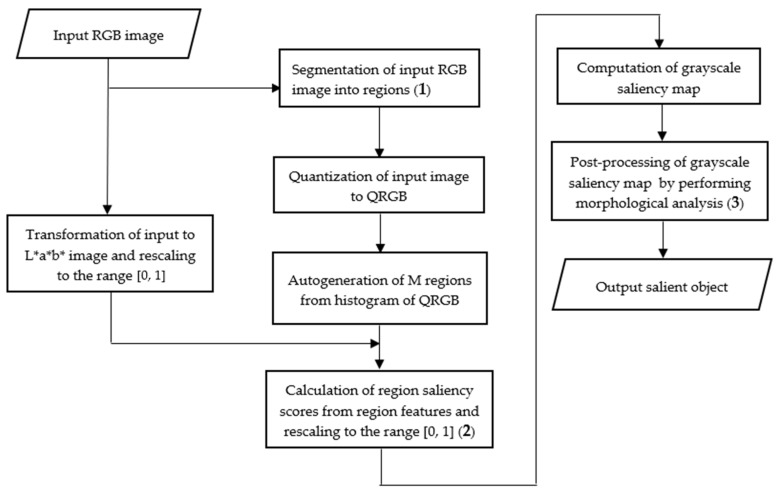
Flowchart of the color histogram clustering method for salient objects detection.

**Figure 2 jimaging-07-00187-f002:**
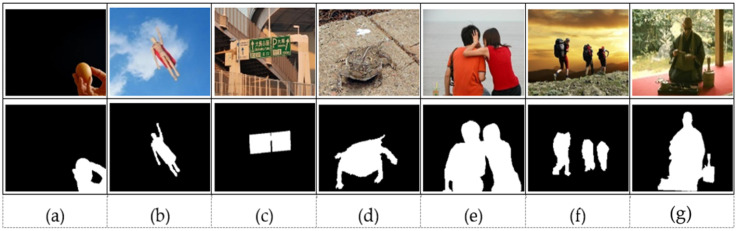
Category of images: (**a**) boundary; (**b**) center; (**c**) complex background; (**d**) low contrast; (**e**) overlap; (**f**) multiple objects; (**g**) ECSSD.

**Figure 3 jimaging-07-00187-f003:**
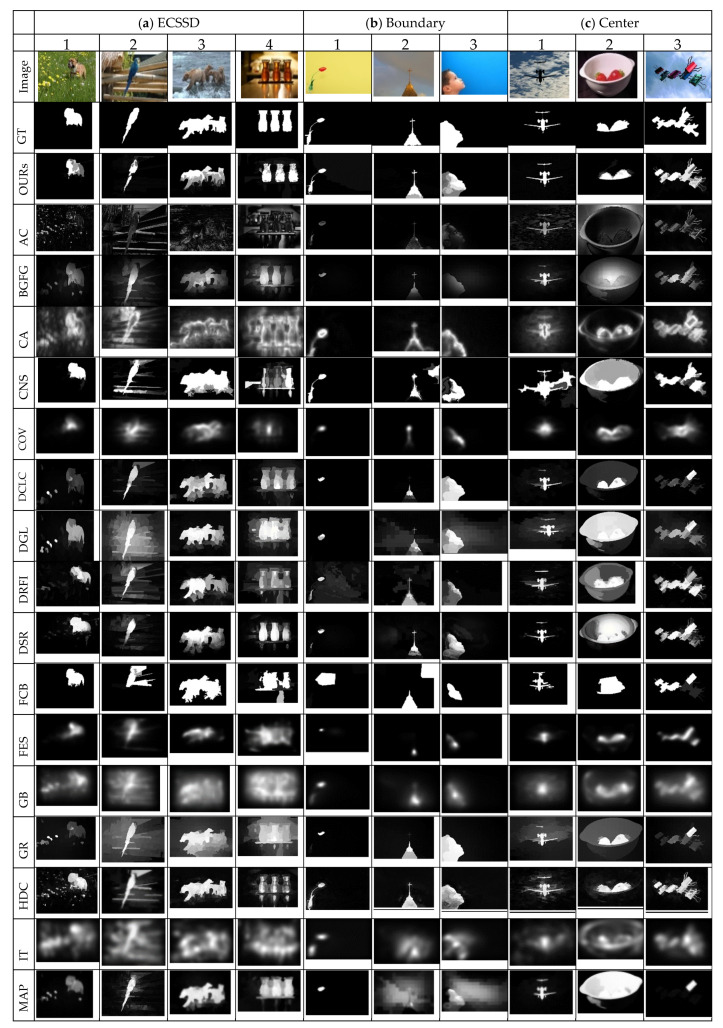
Qualitative performance of the investigated methods on ECSSD and selected categories of images: (**a**) ECSSD; (**b**) Boundary; (**c**) Center; (**d**) Complex Background; (**e**) Low Contrast; (**f**) Multiple Objects; (**g**) Overlap.

**Figure 4 jimaging-07-00187-f004:**
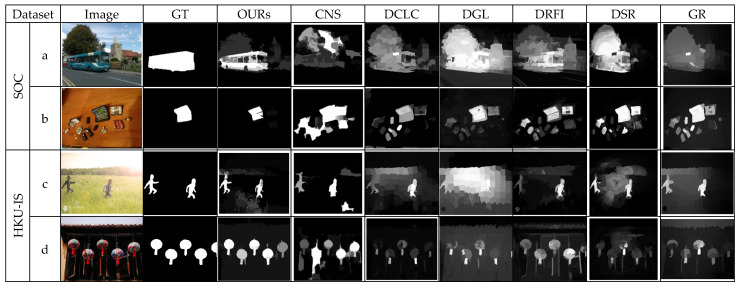
Qualitative performance of the proposed method, CNS, DCLC, DGL, DRFI, DSR and, GR on SOC and HKU-IS datasets: (**a**) Salient object with the heterogeneous background; (**b**) Salient object surrounded by multiple non-salient objects; (**c**) Salient objects with illumination change; (**d**) Multiple salient objects.

**Figure 5 jimaging-07-00187-f005:**
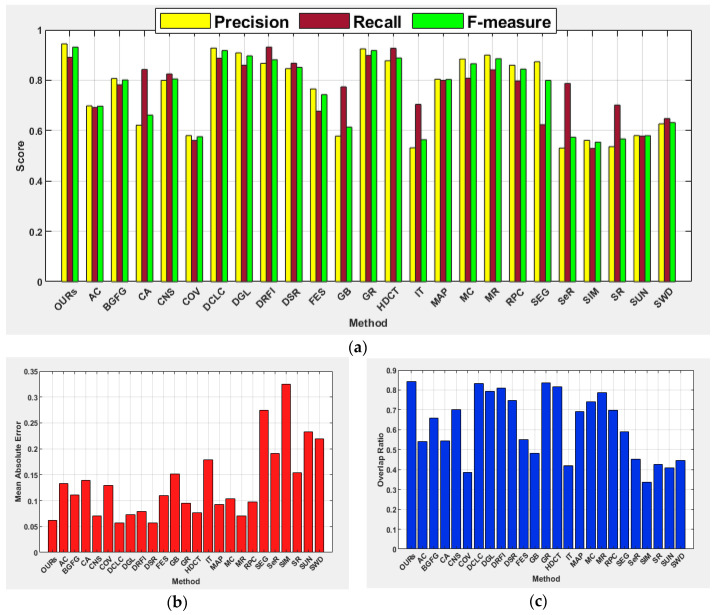
(**a**) F-measure; (**b**) MAR and (**c**) OR on image category: Boundary.

**Figure 6 jimaging-07-00187-f006:**
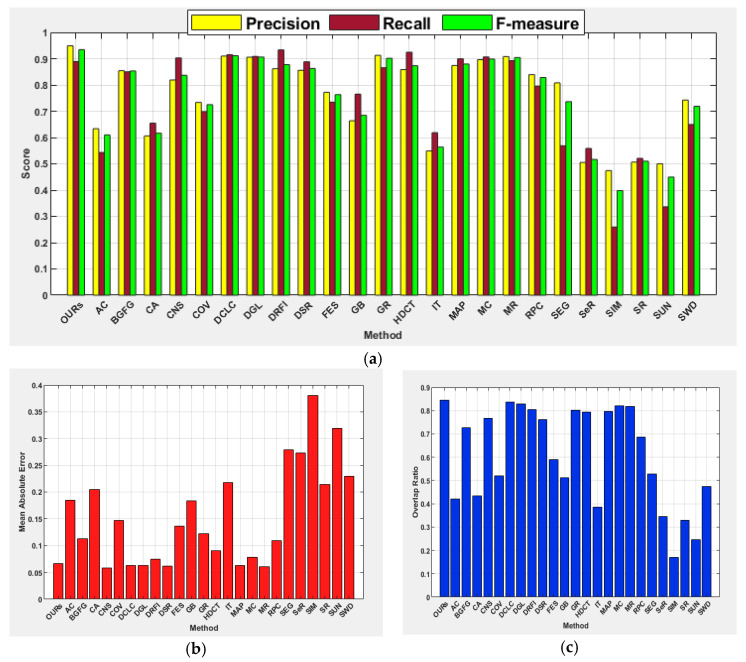
(**a**) F-measure; (**b**) MAR and (**c**) OR on image category: Center.

**Figure 7 jimaging-07-00187-f007:**
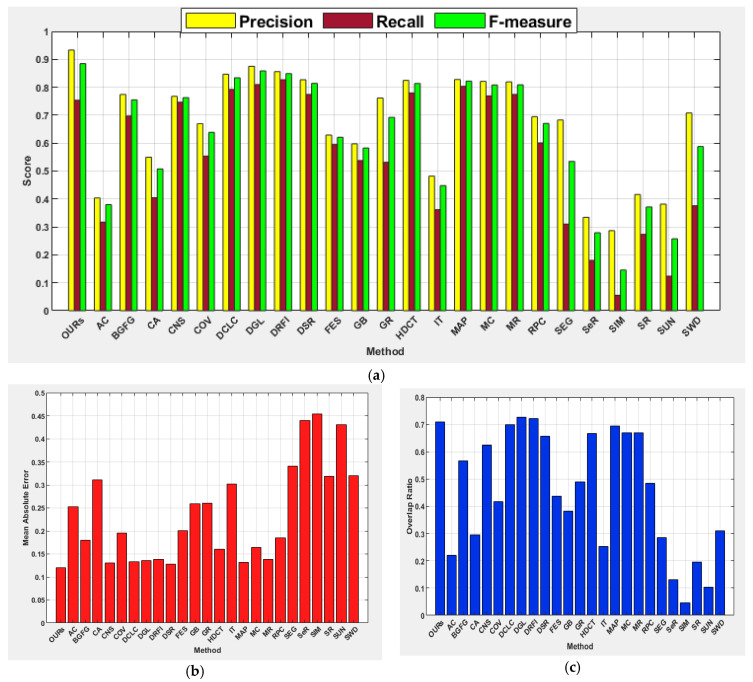
(**a**) F-measure; (**b**) MAE and (**c**) OR on image category: Complex background.

**Figure 8 jimaging-07-00187-f008:**
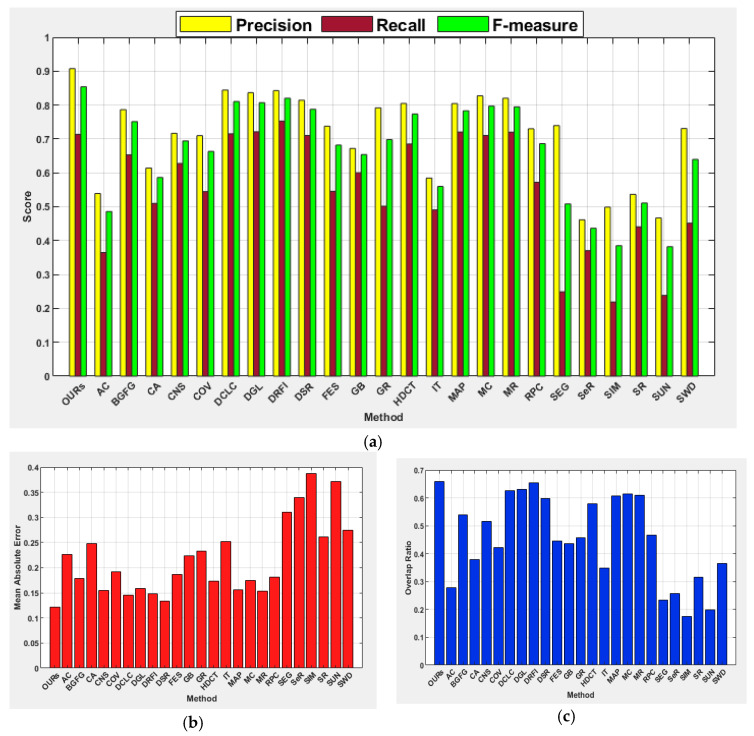
(**a**) F-measure; (**b**) MAE and (**c**) OR on image category: Low contrast.

**Figure 9 jimaging-07-00187-f009:**
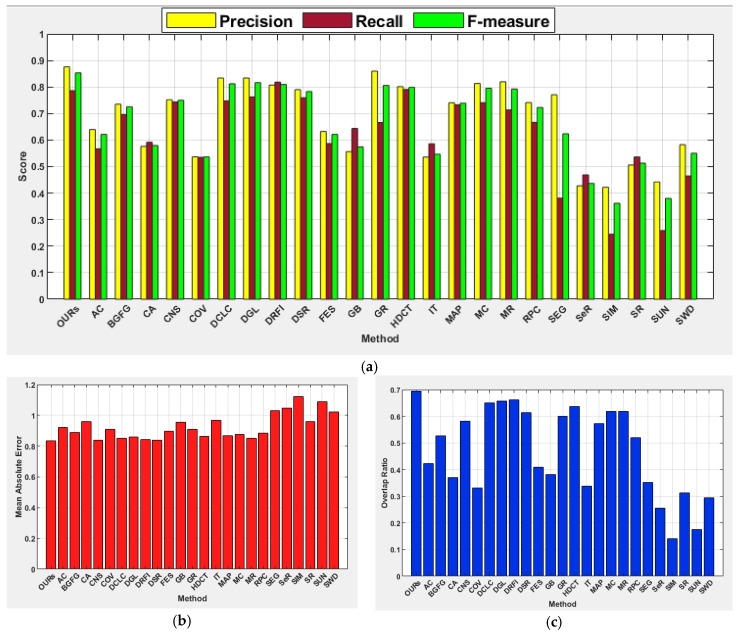
(**a**) F-measure; (**b**) MAE and (**c**) OR on image category: Multiple salient objects.

**Figure 10 jimaging-07-00187-f010:**
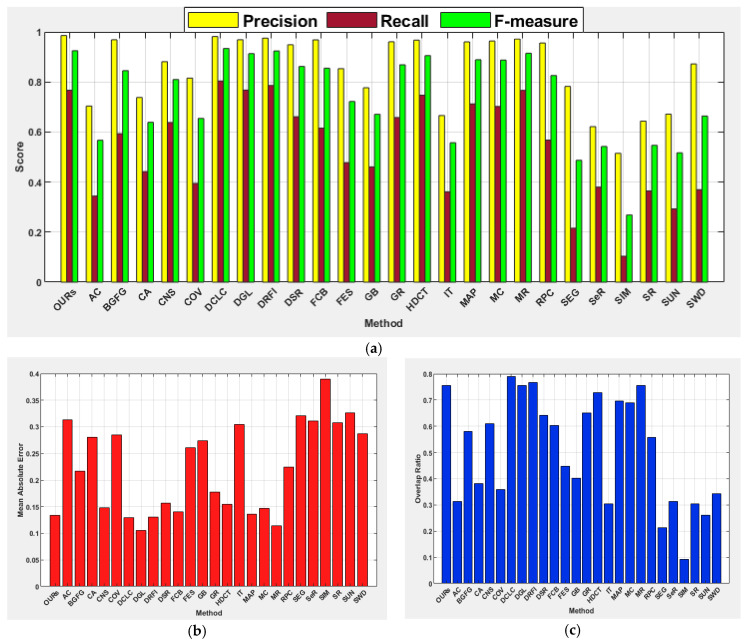
(**a**) F-measure; (**b**) MAE and (**c**) OR on image category: Overlapped objects.

**Figure 11 jimaging-07-00187-f011:**
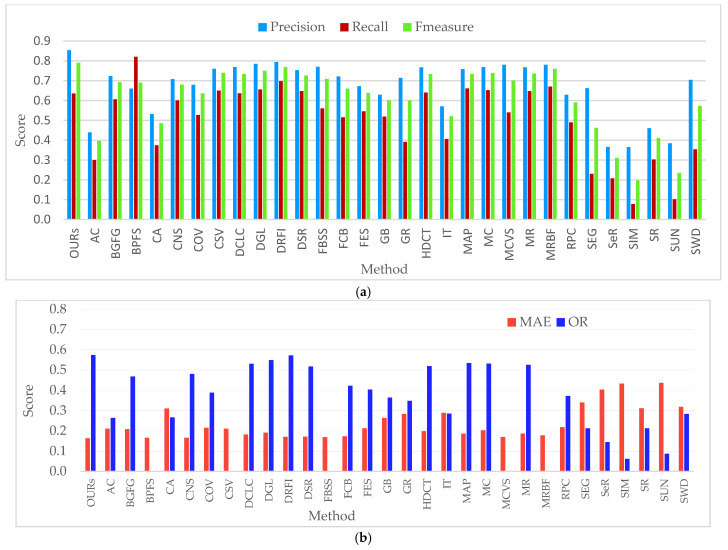
(**a**) F-measure; (**b**) MAE and OR on ECSSD dataset.

**Table 2 jimaging-07-00187-t002:** The performance statistics for six categories of images. The up arrow ↑ indicates that a higher value gives better performance, and the down arrow ↓ shows that a lower value gives better performance.

(a)	Metric	OURs	AC	BGFG	CA	CNS	COV	DCLC	DGL	DRFI	DSR	FES	GB	GR
Boundary (350) ^1^	Precision ↑	0.945	0.698	0.807	0.621	0.800	0.580	0.928	0.909	0.867	0.846	0.765	0.578	0.924
	Recall ↑	0.891	0.692	0.782	0.843	0.825	0.561	0.887	0.859	0.932	0.868	0.677	0.774	0.898
	F-measure ↑	0.932	0.697	0.801	0.661	0.805	0.576	0.918	0.897	0.882	0.851	0.743	0.614	0.918
	MAE ↓	0.062	0.133	0.111	0.140	0.071	0.130	0.057	0.073	0.080	0.057	0.110	0.152	0.095
	OR ↑	0.844	0.541	0.659	0.545	0.700	0.387	0.832	0.794	0.808	0.747	0.551	0.480	0.837
	Metric		HDCT	IT	MAP	MC	MR	RPC	SEG	SeR	SIM	SR	SUN	SWD
	Precision ↑		0.878	0.532	0.804	0.884	0.900	0.860	0.873	0.531	0.562	0.536	0.58	0.627
	Recall ↑		0.927	0.705	0.799	0.808	0.841	0.797	0.624	0.787	0.529	0.701	0.578	0.648
	F-measure ↑		0.888	0.564	0.803	0.865	0.886	0.844	0.799	0.574	0.554	0.567	0.58	0.632
	MAE ↓		0.077	0.179	0.092	0.104	0.070	0.097	0.274	0.191	0.325	0.154	0.233	0.219
	OR ↑		0.816	0.417	0.691	0.741	0.785	0.698	0.588	0.452	0.337	0.425	0.408	0.446
Center (370) ^1^	Metric	OURs	AC	BGFG	CA	CNS	COV	DCLC	DGL	DRFI	DSR	FES	GB	GR
	Precision ↑	0.949	0.633	0.854	0.606	0.819	0.733	0.910	0.906	0.862	0.856	0.772	0.664	0.913
	Recall ↑	0.889	0.543	0.850	0.655	0.903	0.699	0.915	0.909	0.933	0.888	0.734	0.765	0.866
	F-measure ↑	0.934	0.610	0.853	0.617	0.837	0.725	0.911	0.906	0.877	0.863	0.763	0.685	0.901
	MAE ↓	0.067	0.184	0.112	0.204	0.058	0.147	0.063	0.063	0.075	0.062	0.136	0.183	0.122
	OR ↑	0.846	0.420	0.727	0.435	0.768	0.520	0.838	0.830	0.804	0.762	0.589	0.511	0.801
	Metric		HDCT	IT	MAP	MC	MR	RPC	SEG	SeR	SIM	SR	SUN	SWD
	Precision ↑		0.859	0.549	0.874	0.896	0.908	0.839	0.808	0.505	0.474	0.507	0.500	0.742
	Recall ↑		0.925	0.618	0.899	0.906	0.892	0.795	0.568	0.559	0.259	0.521	0.336	0.649
	F-measure ↑		0.873	0.564	0.88	0.898	0.904	0.828	0.736	0.516	0.398	0.510	0.450	0.719
	MAE ↓		0.091	0.218	0.063	0.079	0.061	0.109	0.279	0.273	0.381	0.214	0.319	0.230
	OR ↑		0.794	0.386	0.796	0.82	0.818	0.686	0.527	0.344	0.171	0.330	0.245	0.475
Complex background (210) ^1^	Metric	OURs	AC	BGFG	CA	CNS	COV	DCLC	DGL	DRFI	DSR	FES	GB	GR
	Precision ↑	0.933	0.404	0.774	0.550	0.768	0.670	0.847	0.875	0.856	0.827	0.629	0.598	0.762
	Recall ↑	0.753	0.317	0.697	0.405	0.747	0.554	0.793	0.810	0.827	0.774	0.595	0.537	0.531
	F-measure ↑	0.885	0.380	0.755	0.508	0.763	0.639	0.834	0.859	0.849	0.814	0.621	0.583	0.692
	MAE ↓	0.120	0.253	0.179	0.311	0.130	0.195	0.133	0.135	0.138	0.127	0.200	0.259	0.26
	OR ↑	0.710	0.22	0.568	0.295	0.623	0.418	0.700	0.726	0.721	0.657	0.438	0.384	0.49
	Metric		HDCT	IT	MAP	MC	MR	RPC	SEG	SeR	SIM	SR	SUN	SWD
	Precision ↑		0.824	0.482	0.828	0.821	0.819	0.695	0.683	0.334	0.287	0.416	0.381	0.708
	Recall ↑		0.780	0.362	0.803	0.769	0.774	0.601	0.310	0.180	0.055	0.273	0.123	0.376
	F-measure ↑		0.814	0.448	0.822	0.808	0.809	0.671	0.535	0.279	0.146	0.371	0.257	0.588
	MAE ↓		0.160	0.303	0.131	0.164	0.139	0.185	0.341	0.439	0.454	0.318	0.430	0.321
	OR ↑		0.666	0.254	0.694	0.669	0.670	0.485	0.285	0.131	0.045	0.197	0.102	0.310
Low contrast (165) ^1^	Metric	OURs	AC	BGFG	CA	CNS	COV	DCLC	DGL	DRFI	DSR	FES	GB	GR
	Precision ↑	0.908	0.539	0.787	0.614	0.717	0.710	0.844	0.837	0.843	0.814	0.738	0.672	0.792
	Recall ↑	0.715	0.365	0.653	0.510	0.628	0.545	0.715	0.721	0.753	0.710	0.545	0.600	0.501
	F-measure ↑	0.854	0.486	0.751	0.586	0.694	0.663	0.810	0.807	0.820	0.788	0.682	0.654	0.698
	MAE ↓	0.122	0.227	0.178	0.248	0.155	0.193	0.146	0.159	0.148	0.134	0.187	0.224	0.233
	OR ↑	0.659	0.278	0.538	0.38	0.516	0.423	0.625	0.631	0.654	0.599	0.445	0.436	0.457
	Metric		HDCT	IT	MAP	MC	MR	RPC	SEG	SeR	SIM	SR	SUN	SWD
	Precision ↑		0.805	0.585	0.804	0.827	0.820	0.730	0.740	0.461	0.499	0.537	0.467	0.731
	Recall ↑		0.685	0.491	0.720	0.710	0.720	0.572	0.249	0.370	0.219	0.441	0.238	0.452
	F-measure ↑		0.774	0.560	0.783	0.797	0.795	0.686	0.508	0.437	0.385	0.511	0.382	0.640
	MAE ↓		0.173	0.252	0.156	0.175	0.153	0.182	0.310	0.340	0.388	0.261	0.371	0.274
	OR ↑		0.578	0.348	0.606	0.613	0.61	0.466	0.233	0.257	0.175	0.316	0.198	0.364
Multiple objects (160) ^1^	Metric	OURs	AC	BGFG	CA	CNS	COV	DCLC	DGL	DRFI	DSR	FES	GB	GR
	Precision	0.876	0.640	0.735	0.576	0.752	0.537	0.84	0.834	0.807	0.790	0.633	0.556	0.86
	Recall	0.786	0.567	0.696	0.592	0.743	0.535	0.748	0.762	0.818	0.759	0.587	0.644	0.666
	F-measure	0.853	0.621	0.726	0.580	0.750	0.537	0.812	0.816	0.810	0.783	0.621	0.574	0.806
	MAE ↓	0.836	0.921	0.888	0.958	0.840	0.911	0.850	0.860	0.842	0.839	0.896	0.955	0.909
	OR	0.695	0.425	0.528	0.371	0.582	0.331	0.652	0.656	0.663	0.614	0.410	0.382	0.599
	Metric		HDCT	IT	MAP	MC	MR	RPC	SEG	SeR	SIM	SR	SUN	SWD
	Precision		0.801	0.536	0.741	0.813	0.820	0.741	0.771	0.427	0.422	0.506	0.442	0.583
	Recall		0.791	0.586	0.733	0.741	0.714	0.666	0.381	0.469	0.245	0.537	0.259	0.464
	F-measure		0.799	0.547	0.739	0.795	0.793	0.723	0.624	0.436	0.362	0.513	0.380	0.550
	MAE ↓		0.864	0.967	0.866	0.878	0.851	0.883	1.032	1.047	1.124	0.960	1.091	1.021
	OR		0.638	0.338	0.574	0.619	0.619	0.521	0.352	0.255	0.141	0.314	0.176	0.295
Overlap (250) ^1^	Metric	OURs	AC	BGFG	CA	CNS	COV	DCLC	DGL	DRFI	DSR	FCB	FES	GB
	Precision ↑	0.986	0.703	0.969	0.738	0.881	0.815	0.981	0.969	0.975	0.949	0.968	0.853	0.777
	Recall ↑	0.767	0.344	0.593	0.442	0.638	0.395	0.804	0.767	0.756	0.661	0.615	0.478	0.461
	F-measure ↑	0.925	0.567	0.845	0.639	0.810	0.654	0.934	0.913	0.924	0.862	0.855	0.722	0.671
	MAE ↓	0.134	0.313	0.217	0.280	0.148	0.285	0.130	0.105	0.130	0.157	0.140	0.260	0.274
	OR ↑	0.757	0.313	0.581	0.381	0.609	0.358	0.790	0.755	0.768	0.641	0.603	0.447	0.402
	Metric	GR	HDCT	IT	MAP	MC	MR	RPC	SEG	SeR	SIM	SR	SUN	SWD
	Precision ↑	0.96	0.967	0.666	0.96	0.963	0.971	0.956	0.782	0.622	0.515	0.644	0.671	0.872
	Recall ↑	0.658	0.747	0.361	0.712	0.702	0.767	0.568	0.216	0.38	0.104	0.365	0.293	0.370
	F-measure ↑	0.868	0.906	0.557	0.889	0.887	0.915	0.826	0.487	0.542	0.269	0.547	0.517	0.664
	MAE ↓	0.178	0.154	0.304	0.136	0.147	0.114	0.225	0.321	0.311	0.389	0.307	0.326	0.287
	OR ↑	0.65	0.728	0.305	0.697	0.690	0.755	0.556	0.214	0.312	0.092	0.304	0.260	0.342

^1^ Number of images.

**Table 3 jimaging-07-00187-t003:** Performance statistics on ECSSD dataset in terms of precision, recall, F-measure, MAE and OR.

Method	Precision	Recall	F-measure	MAE	OR	Method	Precision	Recall	F-measure	MAE	OR
OURs	0.853	0.635	0.790	0.163	0.573	GR	0.714	0.391	0.600	0.283	0.348
AC	0.439	0.300	0.396	0.210	0.263	HDCT	0.767	0.640	0.733	0.198	0.519
BGFG	0.723	0.606	0.692	0.208	0.467	IT	0.570	0.406	0.521	0.289	0.285
BPFS	0.660	0.820	0.690	0.166		MAP	0.758	0.661	0.733	0.185	0.534
CA	0.532	0.374	0.485	0.310	0.266	MC	0.768	0.652	0.738	0.202	0.531
CNS	0.708	0.600	0.680	0.166	0.480	MCVS	0.780	0.540	0.700	0.170	
COV	0.679	0.527	0.636	0.215	0.388	MR	0.767	0.647	0.736	0.186	0.525
CSV	0.760	0.650	0.740	0.210		MRBF	0.780	0.670	0.760	0.177	
DCLC	0.769	0.636	0.734	0.182	0.530	RPC	0.629	0.489	0.590	0.218	0.372
DGL	0.785	0.655	0.750	0.191	0.548	SEG	0.662	0.230	0.462	0.340	0.212
DRFI	0.794	0.698	0.769	0.170	0.572	SeR	0.366	0.207	0.311	0.404	0.144
DSR	0.753	0.647	0.726	0.171	0.517	SIM	0.365	0.078	0.197	0.433	0.062
FBSS	0.770	0.560	0.709	0.169		SR	0.460	0.302	0.411	0.311	0.212
FCB	0.721	0.515	0.660	0.173	0.422	SUN	0.384	0.102	0.235	0.437	0.087
FES	0.672	0.545	0.638	0.212	0.404	SWD	0.704	0.354	0.573	0.318	0.283
GB	0.629	0.519	0.600	0.263	0.364						

**Table 4 jimaging-07-00187-t004:** Comparison with deep learning methods in terms of F-measure and MAE on ECSSD dataset.

Method	F-Measure	MAE
MSNSD-A [38]	0.777	0.171
MSNSD [38]	0.774	0.179
DS [92]	0.759	0.160
LCNN [91]	0.715	0.162
[141]	0.430	0.255
TSL [90]	0.737	0.178
MCDL [93]	0.732	
OURs	0.790	0.163

**Table 5 jimaging-07-00187-t005:** Results of precision, recall, F-Measure, MAE and OR on HKU-IS and SOC datasets.

Datasets	HKU-IS					SOC				
Metrics	Precision	Recall	F-Measure	MAE	OR	Precision	Recall	F-Measure	MAE	OR
DCLC	0.724	0.653	0.707	0.160	0.517	0.558	0.499	0.543	0.215	0.236
DGL	0.725	0.672	0.712	0.189	0.528	0.568	0.505	0.552	0.263	0.244
DRFI	0.753	0.755	0.754	0.144	0.577	0.560	0.563	0.561	0.219	0.356
MSNSD-A [38]			0.837	0.071						
MSNSD [38]			0.837	0.071						
MCDL			0.743	0.093						
OURs	0.813	0.673	0.776	0.144	0.578	0.650	0.531	0.618	0.202	0.389

**Table 6 jimaging-07-00187-t006:** Average running time of 25 methods on ECSSD dataset.

**Method**	**OURS**	**AC**	**BGFG**	**CA**	**CNS**	**COV**	**DCLC**	**DGL**	**DRFI**	**DSR**	**FES**	**GB**	**GR**
Time (s)	0.23	80.33	5.56	15.15	11.34	4.29	0.47	1.33	6.16	1.82	0.21	0.52	0.36
**Method**	**HDCT**	**IT**	**MAP**	**MC**	**MR**	**RPC**	**SEG**	**SeR**	**SIM**	**SR**	**SUN**	**SWD**	
Time (s)	4.17	0.26	0.21	0.24	0.54	2.08	1.91	0.51	0.39	0.12	2.39	0.12	

## Data Availability

The ECSSD dataset is available at https://www.cse.cuhk.edu.hk/leojia/projects/hsaliency/dataset.html (Accessed on 26 May 2019). The MSRA10K dataset is available at https://mmcheng.net/msra10k/ (Accessed on 30 May 2019). The SED2 dataset is available at https://www.wisdom.weizmann.ac.il/~vision/Seg_Evaluation_DB/dl.html (Accessed on 30 May 2019). The ASD dataset is available at https://www.epfl.ch/labs/ivrl/research/saliency/frequency-tuned-salient-region-detection/ (Accessed on 30 May 2019). The DUT OMRON dataset is available at http://saliencydetection.net/dut-omron/ (Accessed on 15 January 2020). The ImgSal dataset is available at https://qualinet.github.io/databases/image/imgsal_mcgill_database_for_saliency_detection/ (Accessed on 20 July 2020). The HKU-IS dataset is available at https://i.cs.hku.hk/~yzyu/research/deep_saliency.html (Accessed on 15 January 2021). The SOC dataset is available at http://dpfan.net/SOCBenchmark/ (Accessed on 24 June 2021).

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
