# Peer review of "Detecting Salient Image Objects Using Color Histogram Clustering for Region Granularity"

_2313-433X, 2021, doi:10.3390/jimaging7090187_

Round 1

Reviewer 1 Report

   After reviewing the article with the name "Detecting salient image objects using color histograms clustering for region granularity" I conclude that the results are worthy of being published. Nonetheless, an improvement of the description of the 3 steps that the proposed method used is required. For example equations 8 and 9. I suggest to the authors include an algorithm for a better understanding of the proposed approach.

Also, some parts of the paper are hard to understand due to the redaction style. I propose to produce an improved version of the paper with better English writing. I also found several type errors in the text, some of them are shown below.

Intelligencem; apositely; diversed; contrastas; amomg; esenial; esixting; forground; leaque; differencs; theperformance; Matric

Author Response

Comments from Reviewer 1

“Description of the method must be improved “

Response:

The description of the method has been improved for better understanding.

Nonetheless, an improvement of the description of the 3 steps that the proposed method used is required. For example equations 8 and 9. I suggest to the authors include an algorithm for a better understanding of the proposed approach.”

Response:

Thank you for pointing this out and we agree with this, and all inconsistencies identified in the description of the method have been rectified. For better understanding, the method is introduced with a flowchart.

Also, some parts of the paper are hard to understand due to the redaction style. I propose to produce an improved version of the paper with better English writing. I also found several type errors in the text, some of them are shown below.”

Intelligencem; apositely; diversed; contrastas; amomg; esenial; esixting; forground; leaque; differencs; theperformance; Matric

Response:

Identified sections are paraphrased and all changes are highlighted in red. All typographical errors have been corrected.

Reviewer 2 Report

It would be very interesting to evaluate the proposed method based on the COCO dataset.

Also, in order to clarify the method, it would be very intelligent to introduce it with a flowchart presenting all the steps.

Is there any pre-processing step. The images are denoised in a preprocessing step ?

Author Response

Comments from Reviewer 2

Description of the method can be improved

Response:

The description of the method has been improved for better understanding.

It would be very interesting to evaluate the proposed method based on the COCO dataset.”

Response:

We agree with this suggestion and have incorporated the dataset. Salient objects in clutter (SOC) which is a subset of the COCO dataset have been used to test our method.

Also, in order to clarify the method, it would be very intelligent to introduce it with a flowchart presenting all the steps.”

Response:

Thank you for this suggestion and the method is introduced with a flowchart.

Is there any pre-processing step. The images are denoised in a preprocessing step?

Response:

No preprocessing step has been used in this method. This has been suggested in the paper to be optional.

Reviewer 3 Report

"Detecting salient image objects using color histogram clustering for region granularity"

Authors have articulated the paper technically good and written well from introduction to conclusion of their study and results.

(a) Good English language and grammar through out the paper. (b) Author's contribution seems to be very good with regard to model description, few more data sets if used then that gives more strength to the article, size of feature vectors, methodology and results analysis.  

Author Response

Comments from Reviewer 3  

Authors have articulated the paper technically good and written well from introduction to conclusion of their study and results.”

Response:

Thank you for the words of encouragement. We have further improved the writing quality.

Good English language and grammar throughout the paper

Response:

We sincerely appreciate your comments. We have further improved the writing quality.

Author's contribution seems to be very good with regard to model description, few more data sets if used then that gives more strength to the article, size of feature vectors, methodology and results analysis

Response:

Two more datasets of HKU-IS and SOC have been further used to evaluate our method.